# TGF-β Signalling Regulates Cytokine Production in Inflammatory Cardiac Macrophages during Experimental Autoimmune Myocarditis

**DOI:** 10.3390/ijms25115579

**Published:** 2024-05-21

**Authors:** Karolina Tkacz, Filip Rolski, Monika Stefańska, Kazimierz Węglarczyk, Rafał Szatanek, Maciej Siedlar, Gabriela Kania, Przemysław Błyszczuk

**Affiliations:** 1Department of Clinical Immunology, Jagiellonian University Medical College, 30-663 Cracow, Polandmonika.stefanska@uj.edu.pl (M.S.); kazimierz.weglarczyk@uj.edu.pl (K.W.); maciej.siedlar@uj.edu.pl (M.S.); 2Center of Experimental Rheumatology, Department of Rheumatology, University Hospital Zurich, University of Zurich, 8952 Schlieren, Switzerland

**Keywords:** TGF-β, inflammatory macrophages, experimental autoimmune myocarditis, cytokines

## Abstract

Myocarditis is characterized by an influx of inflammatory cells, predominantly of myeloid lineage. The progression of myocarditis to a dilated cardiomyopathy is markedly influenced by TGF-β signalling. Here, we investigate the role of TGF-β signalling in inflammatory cardiac macrophages in the development of myocarditis and post-inflammatory fibrosis. Experimental autoimmune myocarditis (EAM) was induced in the *LysM-Cre* × *R26-stop-EYFP* × *Tgfbr2-fl/fl* transgenic mice showing impaired TGF-β signalling in the myeloid lineage and the *LysM-Cre* × *R26-stop-EYFP* control mice. In EAM, immunization led to acute myocarditis on day 21, followed by cardiac fibrosis on day 40. Both strains showed a similar severity of myocarditis and the extent of cardiac fibrosis. On day 21 of EAM, an increase in cardiac inflammatory macrophages was observed in both strains. These cells were sorted and analysed for differential gene expression using whole-genome transcriptomics. The analysis revealed activation and regulation of the inflammatory response, particularly the production of both pro-inflammatory and anti-inflammatory cytokines and cytokine receptors as TGF-β-dependent processes. The analysis of selected cytokines produced by bone marrow-derived macrophages confirmed their suppressed secretion. In conclusion, our findings highlight the regulatory role of TGF-β signalling in cytokine production within inflammatory cardiac macrophages during myocarditis.

## 1. Introduction

Myocarditis represents an acute inflammation affecting the heart and can lead to life-threatening complications, such as heart failure, arrhythmias, or sudden cardiac arrest [1]. As it advances to the chronic stage, it can lead to dilated cardiomyopathy (DCM) in certain patients. DCM is characterized by an imbalance and excessive accumulation of extracellular matrix (ECM) proteins, resulting in cardiac fibrosis, ventricular stiffening, heart enlargement, and functional impairment [2]. Myocarditis may be triggered by a range of infectious agents, including viruses, bacteria, fungi, and protozoa. Moreover, non-infectious factors such as toxins, certain drugs, or autoimmune diseases can also contribute to its onset [3]. Experimental autoimmune myocarditis (EAM) serves as an animal model for investigating the molecular and cellular mechanisms involved in non-infectious myocarditis and post-inflammatory DCM [4]. This model exhibits well-characterized phases of disease progression. In the classical EAM model, induction through immunization with the cardiac-specific alpha myosin heavy chain (α-MyHC) peptide and complete Freund’s adjuvant (CFA) results in an acute myocarditis phase occurring 16–21 days after the initial immunization. In this model, myocarditis is mediated by α-MyHC-reactive CD4+ T cells. In the acute phase of myocarditis, the heart experiences infiltration by inflammatory monocytes, macrophages, granulocytes, as well as T and B lymphocytes. These cells produce pro-inflammatory and profibrotic cytokines and chemokines, exerting an impact on the functioning of other cells. Subsequently, after the resolution of inflammation, some mice progress to develop cardiac dysfunction associated with cardiac fibrosis and ventricular enlargement [5]. Furthermore, a number of experimental animal models of infectious and non-infectious myocarditis were developed, including coxsackievirus B3 passage, T. cruzi infection, PD-1/PD-L1-deficiency, adoptive transfer of dendritic cells loaded with cardiac antigens, transgenic T cell receptor mice, and desmocollin-2 overexpression [5,6,7,8].

Macrophages play a crucial role in maintaining tissue homeostasis and immune responses, tissue repair, debris removal, and modulation of inflammatory processes. Macrophages exhibit a spectrum of functional states with specific functions and responses to stimuli, and at least two major subtypes can be distinguished [9,10]. M1 macrophages are considered a pro-inflammatory phenotype, and their primary function in the heart is to eliminate pathogens, clear cellular debris, and contribute to the regulation of the immune response in the affected cardiac tissue. However, their role extends beyond immune surveillance, as they actively secrete various pro-inflammatory cytokines and chemokines such as IL-1, IL-12, IL-23, TNF-α, and reactive oxygen species that influence the behaviour of neighbouring cells, including other immune cells, fibroblasts, and cardiomyocytes. An excessive or prolonged M1 response can contribute to tissue damage and the exacerbation of inflammation. M2 macrophages, on the other hand, are linked to anti-inflammatory and tissue repair functions. As the inflammatory response in myocarditis progresses, there is a transition towards an M2 phenotype. M2 macrophages release anti-inflammatory cytokines, including IL-10 and TGF-β, promoting resolution of inflammation and tissue fibrosis (Figure 1) [11].

TGF-β is a widely distributed pleiotropic cytokine that regulates numerous cellular processes, including proliferation, differentiation, cytoskeletal reorganization, and the synthesis of ECM proteins. TGF-β exists in three isoforms: TGF-β1, -β2, and -β3, all of which bind to the same receptors. Upon proteolytic activation of a latent form, active TGF-β binds to the transmembrane TGF-β type II receptor (TGFβR2), recruiting and activating the TGF-β type I receptor. This activation initiates signal transduction through both a canonical Smad-dependent pathway and several Smad-independent signalling pathways [12]. TGF-β is acknowledged as a master regulator of cardiac fibrogenesis [13]. Mechanistically, TGF-β signalling dysregulates the expression of target genes that control cell proliferation and the production of structural and ECM proteins, primarily within cardiac fibroblasts. Despite the well-studied role of TGF-β in cardiac fibroblasts, its influence on inflammatory macrophages in the context of inflammatory heart disease remains unclear. In this manuscript, we aimed to investigate the role of TGF-β signalling in regulation of inflammatory and fibrotic processes within the subset of inflammatory macrophages in the context of EAM. 

## 2. Results

To assess the impact of TGF-β signalling on cardiac macrophages in inflammatory heart disease, we utilized the EAM model and transgenic mice with defective TGF-β receptor signalling specifically in the myeloid lineage. The transgenic construct *LysM-Cre* × *R26-stop-EYFP* × *Tgfbr2-fl/fl* was employed for the identification of myeloid cells lacking TGF-β receptor 2 (TGFBR2) through the expression of the reporter protein EYFP. The use of EYFP is crucial because only approximately half of heart inflammatory CD11b+ myeloid cells expressed Cre recombinase in this construct. *LysM-Cre* × *R26-stop-EYFP* mice with intact TGFBR2 served as controls for our study. Immunization of mice with α-MyHC and CFA led to acute myocarditis on day 21, followed by cardiac fibrosis on day 40 (Figure 2A). There were no statistically significant differences in the severity of myocarditis and the extent of cardiac fibrosis observed between the two mouse strains (Figure 2B,C).

In the next step, we characterized EYFP-positive cells in the hearts both at baseline (day 0) and during the acute phase of myocarditis (day 21). Cardiac EYFP-positive cells expressed the pan-myeloid marker CD11b, and many of them were positive for macrophage markers CD64 and MerTK (Figure 2A). Notably, inflammatory macrophages are recognized for their high expression of Ly6c [14]. In our model, we observed an increase in Ly6c^hi^ cells within the EYFP-positive population on day 21 of EAM (Figure 3A). Consequently, we defined EYFP^+^CD11b^+^CD64^+^MerTK^+^Ly6c^hi^ cells present in the hearts on day 21 of EAM as cardiac inflammatory macrophages. It is important to note that both the number of EYFP-positive cells and the population of EYFP^+^CD11b^+^CD64^+^MerTK^+^Ly6c^hi^ inflammatory macrophages were significantly higher in inflamed hearts (day 21) compared to the baseline (day 0) in both mouse strains (Figure 3B). Moreover, the efficiency of Cre recombinase presented as the percentage of EYFP+ cells within CD11b^+^ cells was similar in both strains on d0 and d21 (Figure 3B).

On day 21 of EAM, we sorted EYFP^+^CD11b^+^CD64^+^MerTK^+^Ly6c^hi^ inflammatory macrophages from the hearts of both *LysM-Cre* × *R26-stop-EYFP* × *Tgfbr2-fl/fl* and *LysM-Cre* × *R26-stop-EYFP* mice and subsequently performed whole-genome transcriptomics (Figure 4A,B). The analysis showed expression of 15’901 genes in at least 4 out of 8 analysed samples. The results of the differential expression analysis revealed an increased expression of 71 genes and a reduced expression of 111 genes in TGFBR2-deficient cells. Gene ontology analysis of differentially regulated genes pointed to processes associated with inflammatory response (GO:0006954) and regulation of cytokine production (GO:0001817) (Figure 4C). The reduced expression of genes encoding inflammatory cytokines and chemokines (*Ifnb1*, *Il23a*, *Il10*, *Il12b*, *Cxcl1*, *Tnf*) and cytokine receptors (*Cx3cr1*, *Ccr4*) in TGFβR2-deficient macrophages was particularly remarkable, as it points to TGF-β-dependent regulation of certain important immune mediators during the inflammatory response in the heart (Figure 4C). 

To confirm TGF-β-dependent production of the identified cytokines and chemokines, we utilized a model of bone marrow-derived macrophages (bmMFs). Approximately half of bmMFs obtained from both mouse strains expressed EYFP, CD11b, CD64, and MerTK (Figure 5A). Sorted EYFP-positive bmMFs were stimulated with LPS and TGF-β, and the secreted cytokines were measured using ELISA. Activated *LysM-Cre* × *R26-stop-EYFP* × *Tgfbr2-fl/fl* bmMFs were observed to produce significantly less CXCL1, IFN-β, TNF-α, and IL-23 compared to *LysM-Cre* × *R26-stop-EYFP* control cells (Figure 5B). These findings confirm that TGF-β signalling regulates the production of selected cytokines in macrophages.

## 3. Discussion

Our previous study demonstrated that TGF-β signalling contributes to the progression of cardiac fibrosis in an EAM model [15,16]. While the profibrotic influence of TGF-β on cardiac fibroblasts is well established, its effects on inflammatory macrophages have remained unclear. In this brief report, we demonstrated the influence of TGF-β signalling on the production of specific, mostly pro-inflammatory, cytokines by inflammatory macrophages in myocarditis. TGF-β is recognized for its role in promoting the transition of macrophages towards an anti-inflammatory state [17]. These anti-inflammatory macrophages play crucial roles in tissue repair, inflammation resolution, and immune modulation. In our study, we focused on analysing inflammatory macrophages during the acute phase of myocarditis. Therefore, the detection of numerous pro-inflammatory cytokines in our analysis was expected. Published studies have shown that under certain circumstances TGF-β can induce pro-inflammatory cytokines, such as IL-1β, TNF-α, and IL-6, in cells of myeloid lineage [18,19,20,21]. Our data similarly revealed a pro-inflammatory effect of TGF-β signalling in heart inflammatory macrophages. While TGF-β did regulate the expression of pro-inflammatory genes in our model, we did not observe a clear TGF-β-dependent shift towards an anti-inflammatory M2 phenotype. This observation could be attributed to the predominant pro-inflammatory signalling in the inflamed cardiac tissue at this stage, suggesting that the inflammatory macrophages may not have initiated the transition from the M1 to M2 phenotype yet. Alternatively, myocarditis in the α-MyHC/CFA EAM model may reflect a microenvironment characteristic of an early stage of inflammation. Typically, in this model, inflammation onset occurs around day 14, followed by acute myocarditis on day 21, and subsequent resolution of inflammation and the onset of fibrosis shortly thereafter. Additionally, the turnover rate of inflammatory macrophages in this model remains unknown. Hence, it is plausible that due to the relatively brief presence of inflammatory macrophages in the heart during myocarditis, this EAM model may lack the typical conversion of pro-inflammatory M1 to anti-inflammatory M2 macrophages regulated by TGF-β signalling. In certain models, this conversion is believed to ameliorate inflammation and trigger fibrosis [22]. It seems that in the EAM model, both the resolution of inflammation and the development of post-inflammatory fibrosis are regulated by other cell types, such as CD4+ T cells. For example, it has been shown that heart-reactive CD4+ T cells play a dominant role in controlling EAM development, and IL-17A produced by Th17 cells promotes fibrosis in this model [23]. Moreover, different types of cells, such as Th1 or Th2, can regulate cytokine production during myocarditis, thereby controlling its resolution or progression or compensating for myeloid cell activity [24]. In line with this hypothesis, our data showed that the development of post-inflammatory fibrosis was not affected in mice with defective TGF-β signalling in the macrophage lineage.

It is important to note the limitations of the genetic construct used in our study. While the *LysM-Cre* transgene is commonly employed for investigating macrophage biology and exhibits high expression and activity of Cre recombinase in macrophages, particularly in mice on the C57BL/6 background [25], our findings revealed only 40–60% efficiency in mice on the BALB/c background. Using a reporter construct such as *R26-stop-EYFP* and cell sorting allows the isolation of cells with confirmed Cre activity for ex vivo analysis. However, it is possible that any potential effect on the cardiac phenotype *LysM-Cre* × *R26-stop-EYFP* × *Tgfbr2-fl/fl* mice could be hidden by the presence of a substantial number of macrophages lacking Cre activity.

Currently, animal models constitute the basis for preclinical studies and can provide the knowledge necessary for the development of translational medicine. The results obtained can be potentially translated into human inflammatory heart diseases as the EAM model mimics the course of the disease, from the acute phase to chronic, progressive cardiomyopathy. Novel therapies targeting TGF-β signalling in various cardiac disorders are under development; therefore, these data may be a valuable source to evaluate their potential under inflammatory conditions in the heart. However, it is important to point out limitations associated with different immunological mechanisms in mice and humans, as well as identical genetic backgrounds in mice, which can likely cause identical responses to the disease-causing agent, while the progression of disease in humans strongly depends on genetic diversity. 

In summary, our findings shed light on the role of TGF-β signalling in regulating the expression of specific pro-inflammatory genes in cardiac inflammatory macrophages during myocarditis. Additionally, our findings provide insights into the inflammatory process within the EAM model. This study thus contributes to a deeper understanding of the pathophysiology of TGF-β in inflammatory heart diseases and furthers our understanding of the EAM model’s characteristics.

## 4. Materials and Methods

### 4.1. Mice 

*LysM-Cre*, *R26-stop-EYFP*, and *Tgfbr2-fl/fl* mice were originally obtained from the Jackson Laboratory and subsequently back-crossed for a minimum of 10 generations on the BALB/c background. The breeding strategy involved crossing *LysM-Cre*, *R26-stop-EYFP*, and *Tgfbr2-fl/fl* mice to generate *LysM-Cre* × *R26-stop-EYFP* and *LysM-Cre* × *R26-stop-EYFP* × *Tgfbr2-fl/fl* mice. The mice were bred under standard conditions, including a 12/12 h light/dark cycle, a room temperature of 20–22 °C, humidity levels of 45–55%, and ad libitum access to food and water. All experiments were conducted in compliance with Polish law and received approval from local authorities (license 489/2021 and 207/2017). Animal experiments adhered to the guidelines outlined in Directive 2010/63/EU of the European Parliament for the protection of animals used for scientific purposes.

### 4.2. Experimental Autoimmune Myocarditis

EAM was induced in 6–8-week-old mice of *LysM-Cre* × *R26-stop-EYFP* and *LysM-Cre* × *R26-stop-EYFP* × *Tgfbr2-fl/fl* genotypes by subcutaneous injection of 200 µg of α-MyHC_614–634_ peptide (Ac-RSLKLMATLFSTYASADR-OH, Caslo, Lyngby, Denmark) emulsified in a 1:1 ratio with complete Freund’s adjuvant (CFA, BD Difco, Franklin Lakes, NJ, USA) on days 0 and 7. At the end of the experiment, on days 21 or 40, the mice were euthanized by cervical dislocation.

### 4.3. Histology

Mouse heart tissues were collected at day 21 or 40 of EAM, fixed for 72 h in 4% formalin, and embedded in paraffin. Myocarditis severity was assessed using Hematoxylin/Eosin sections at day 21 of EAM and graded from 0 to 4, as described previously [26]. Cardiac fibrosis was evaluated at day 40 using conventional Masson’s trichrome staining, measuring the percentage of fibrotic area relative to the total heart area. Slides were analysed with an Olympus BX51 microscope (Olympus, Tokyo, Japan) with 4× lenses, ImageJ software (Version 1.52a, NIH, Bethesda, MA, USA), and custom-made plug-ins.

### 4.4. Cell Sorting 

In order to characterize the phenotype of cardiac macrophages and isolate the subset of inflammatory cardiac macrophages, we collected the mouse hearts on day 21 and performed cell sorting. The collected hearts were perfused with PBS, mechanically dissected, and enzymatically digested with Liberase solution (Roche, Basel, Switzerland) for 45–60 min at 37 °C, and then filtered through 70 µm and 40 µm cell strainers. The cells were suspended in a flow cytometry buffer (2% FBS, 1 mM EDTA in PBS) and treated with an Fc receptor blocker (anti-CD16/32 antibody, 1:100, clone number 93, BioLegend, San Diego, CA, USA) for 15 min at 4 °C. Subsequently, the cells were incubated for 30 min on ice with a combination of fluorochrome-conjugated antibodies: anti-CD11b-APC (1:600, M1/70, Invitrogen, Waltham, MA, USA), anti-CD64-PE (1:300, X54–5/7.1, BioLegend, San Diego, CA, USA), anti-MerTK (1:300, 2B10C42, BioLegend, San Diego, CA, USA), anti-Ly6C-BV (1:300, HK1.4, BioLegend, San Diego, CA, USA). Then, they were washed in 2 mL of flow cytometry buffer, centrifuged at 500× *g* for 5 min, and resuspended in 100 µL flow cytometry buffer. Cell viability was assessed using propidium iodide (Invitrogen, Waltham, MA, USA) added in the amount 1:100. Inflammatory cardiac macrophages were sorted from cell suspensions using FACSAria III (BD Biosciences, San Jose, CA, USA). The data were analysed using BD FACSDiva™ software (version 6.1.2). 

### 4.5. Bulk RNA Sequencing

In order to investigate how TGF-β regulates inflammatory processes in inflammatory macrophages during acute myocarditis, RNA sequencing was performed on RNA isolated from these cells sorted from *LysM-Cre* × *R26-stop-EYFP* and *LysM-Cre* × *R26-stop-EYFP* × *Tgfbr2-fl/fl* mice on day 21 of EAM using total genomic DNA isolation kits (Norgen Biotek Corp., Thorold, ON, Canada) following the manufacturer’s protocol. Each sample represented 20–90,000 cells sorted from one mouse heart. The subsequent steps of library preparation and sequencing were outsourced to an external service (Genewiz, Leipzig, Germany). Briefly, the mRNA samples were initially fragmented, and double-stranded cDNA synthesis was performed. The 5′ end of the cDNA was phosphorylated, and a polyA tail was attached to the 3′ end for PolyA selection rRNA removal. The samples were sequenced on Illumina NovaSeq using 2 × 150 bp sequencing mode, and 20–30 million reads per sample. The sequence reads were quality-assessed, and the remaining adapter sequences were removed using Trimmomatic v.0.36 and then mapped to the GRCm38 reference genome of the house mouse Mus musculus using the STAR aligner v.2.5.2b. The read arrays were then generated using the Subread package v.1.5.2 software. A comparison of gene expression between the groups was performed using DESeq2 (version 1.44). The Wald test was used to generate *p*-values and log_2_fold changes. Differentially expressed genes were considered as those for which the *p*-value was less than 0.05 (*p* < 0.05) and with a log2-fold change > 1 (minimum 2-fold increase). Gene expression levels were grouped on a graph based on normalized counts for each sample and differentially expressed genes in the form of the so-called heatmaps using online software https://software.broadinstitute.org/morpheus/ (accessed on 5 April 2022). Hierarchical clustering by rows and columns was performed using the one minus Pearson correlation method. 

### 4.6. Bone Marrow-Derived Macrophages

Bone marrow cells were obtained by perfusing femurs and tibias from *LysM-Cre* × *R26-stop-EYFP* × *Tgfbr2-fl/fl* and *LysM-Cre* × *R26-stop-EYFP* mice. The bone marrow cells were plated in Petri dishes in DMEM supplemented with 4.5 g/L glucose, 10% FBS, and 50 ng/mL M-CSF and cultured for 10 days under standard conditions—37 °C, 5% CO_2_, and high humidity—to generate bone marrow-derived macrophages. Next, EYFP^+^CD11b^+^CD64^+^MerTK^+^ cells were sorted and seeded into 96-well plates at a density of 20’000 cells per well. The cells were then stimulated with 1 ng/mL LPS (Sigma-Aldrich, Saint Louis, MI, USA) and/or 10 ng/mL TGF-β (PeproTech, London, UK) for 24 h, and the supernatants were analysed using enzyme-linked immunosorbent assay (ELISA).

### 4.7. Enzyme-Linked Immunosorbent Assay

ELISA was used to quantify the concentrations of selected cytokines in vitro in cell supernatants, using commercially available kits for mouse CXCL1 (Biotechne, Minneapolis, MN, USA), IFN-β (Biotechne), IL23 (Biolegend), and TNF-α (Biolegend). The measurements were performed following the manufacturer’s protocols.

### 4.8. Statistical Analysis 

For normally distributed data, we utilized Student’s *t*-test for parametric data and the Kruskal–Wallis test for nonparametric data. Differences were considered statistically significant for *p*  <  0.05. Analyses were performed using GraphPad Prism 8 software (version 8.0.1). All experimental results are presented as mean ± SEM.

## Figures and Tables

**Figure 1 ijms-25-05579-f001:**
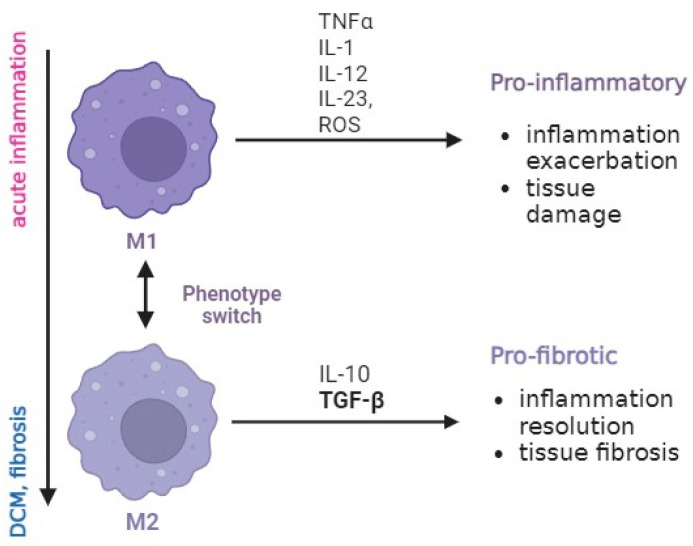
Macrophage phenotypes and their functions in the context of cardiac diseases. The scheme presents M1 (pro-inflammatory) and M2 (pro-fibrotic) macrophage phenotypes. During acute myocarditis, M1 macrophages by secreting TNF-α, IL-1, IL-12, and IL-23 contribute to exacerbation of inflammation and tissue damage. Persistent inflammatory response leads to transition towards an M2 phenotype, which secretes IL-10 and TGF-β, promoting resolution of inflammation and tissue fibrosis.

**Figure 2 ijms-25-05579-f002:**
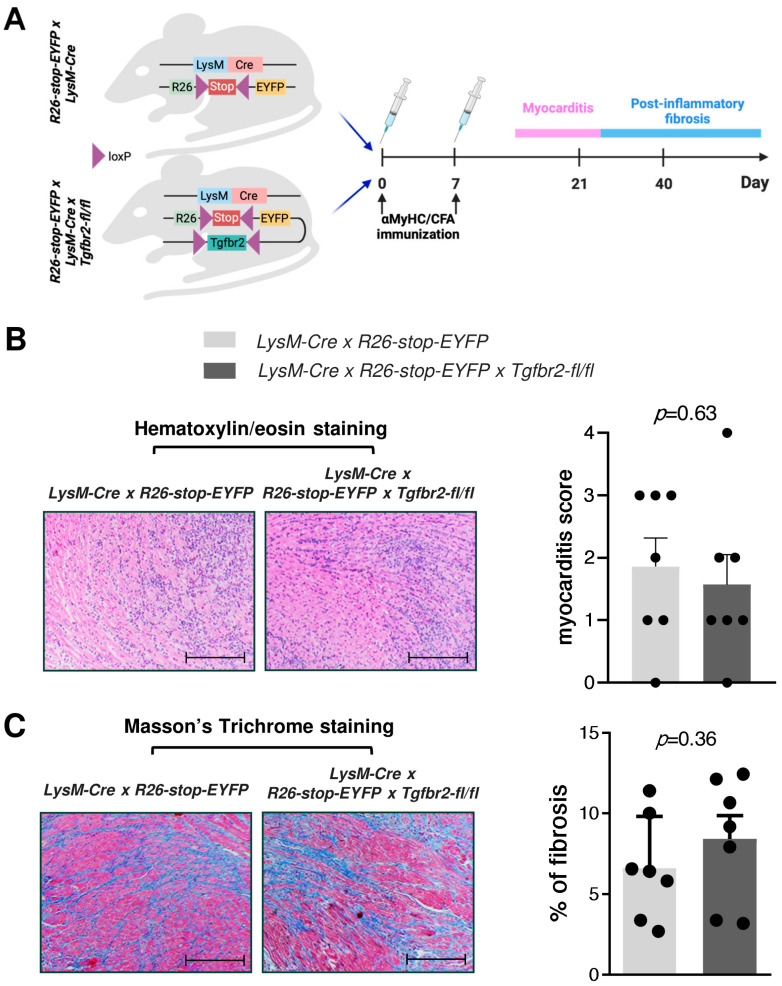
Myocarditis severity and cardiac fibrosis in mice with defective TGF-β signalling in myeloid cells. In panel (**A**), a schematic presentation of the experimental setup is shown. Panel (**B**) displays representative hematoxylin/eosin-stained heart tissue sections on day 21 of EAM, along with the quantification of myocarditis severity. Panel (**C**) shows representative Masson’s trichrome staining of heart tissue sections on day 40, accompanied by the quantification of the fibrotic area. The scale bar is set at 200 μm, 40× magnification. Each dot represents data for one mouse, and the bars present mean values ± SEM. The *p*-values were calculated using the Kruskal–Wallis test (**B**) or unpaired Student’s *t*-test (**C**).

**Figure 3 ijms-25-05579-f003:**
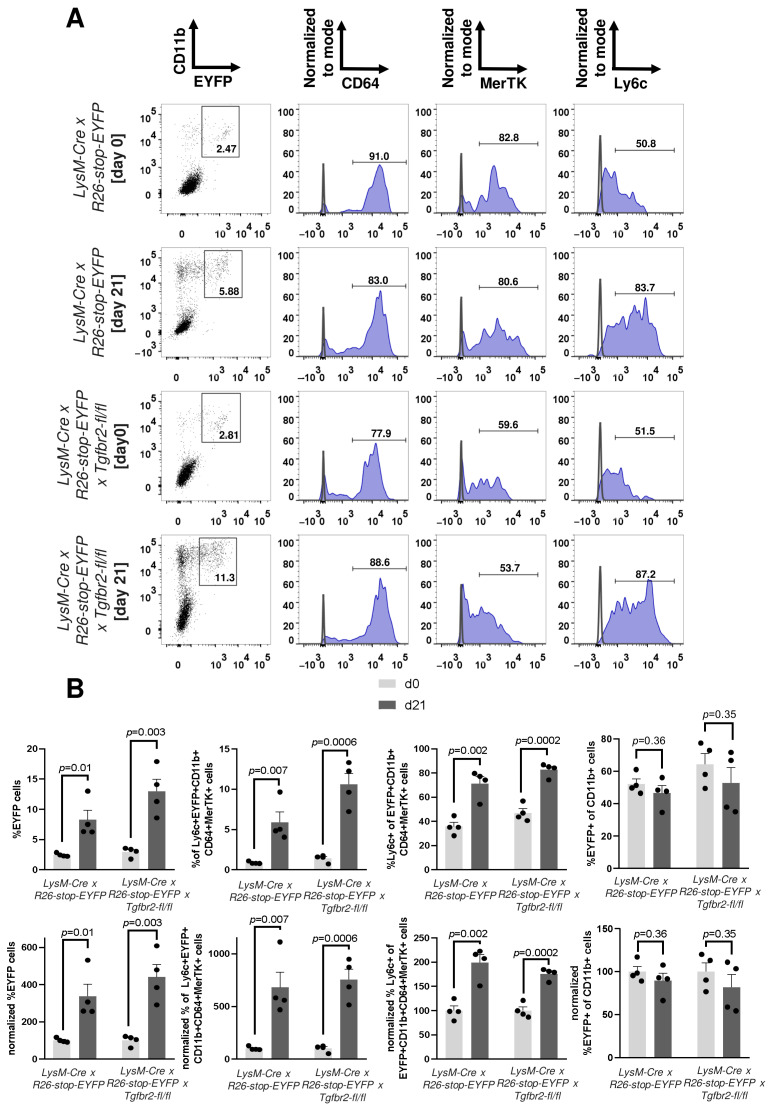
Characteristics of macrophages in myocarditis. In panel (**A**), representative flow cytometry analyses of the indicated antigens on single-cell suspensions from mouse hearts on days 0 and 21 of EAM are shown. Histograms illustrate signals for the respective antigens on gated CD11b^+^EYFP^+^ cells. Panel (**B**) presents quantifications of the percentage and fold change of EYFP^+^ cells, EYFP^+^CD11b^+^CD64^+^MerTK^+^Ly6c^hi^ inflammatory macrophages, Ly6c^hi^ of EYFP^+^CD11b^+^CD64^+^MerTK^+^ macrophages, and EYFP^+^ of CD11b^+^ cells on days 0 and 21 of EAM. Each dot represents data for one heart, and the bars represent mean values ± SEM. The *p*-values were calculated using the unpaired Student’s *t*-test.

**Figure 4 ijms-25-05579-f004:**
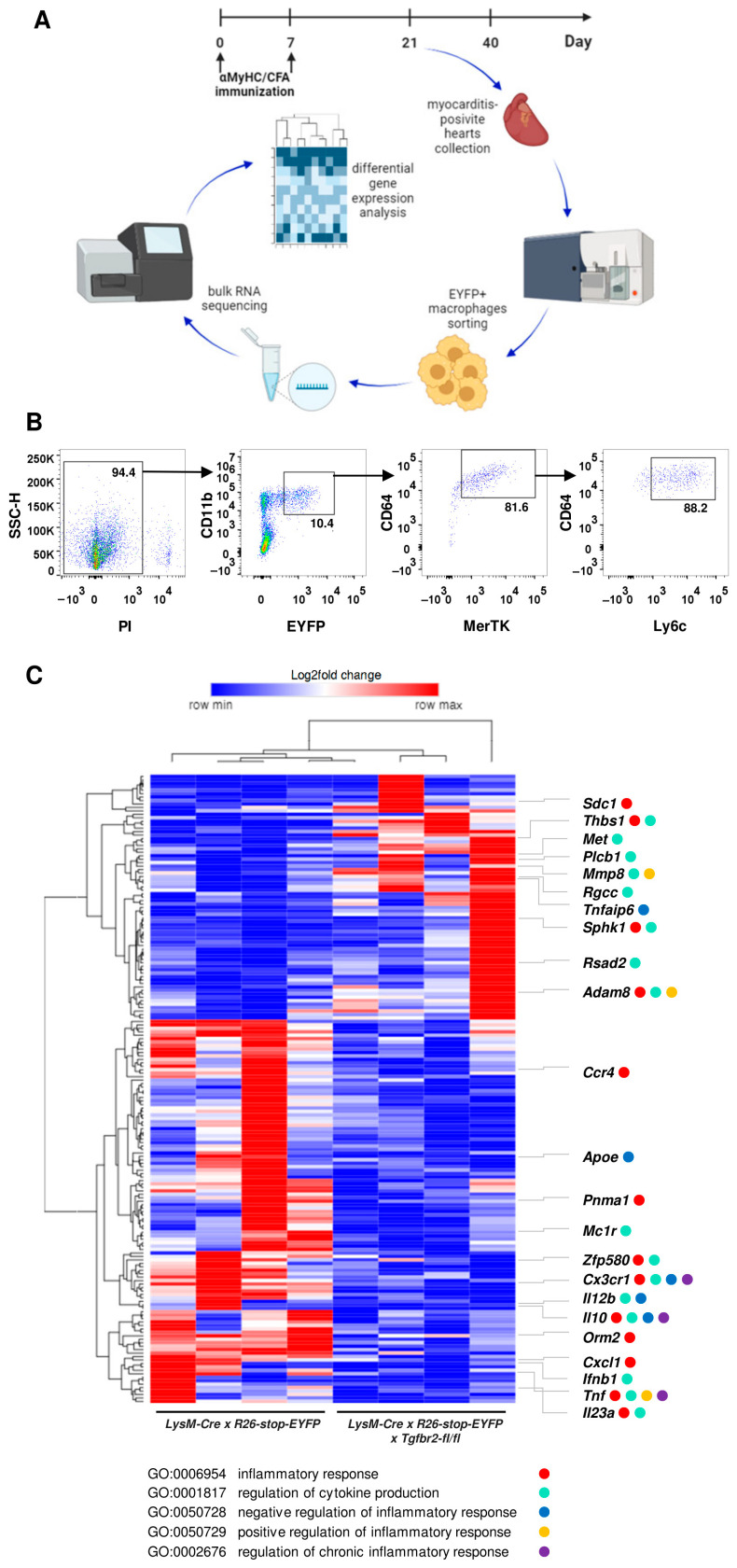
Bulk RNAseq on inflammatory macrophages in myocarditis. In panel (**A**), a schematic of the experimental setup is presented. Panel (**B**) displays the gating strategy for sorting EYFP^+^CD11b^+^CD64^+^MerTK^+^Ly6c^hi^ heart inflammatory macrophages on day 21 of EAM. Panel (**C**) shows a heat map of differentially expressed genes between the indicated groups with log_2_fold change > 1 (fold change > 2) and *p*-value < 0.05. Hierarchical clustering by rows and columns was performed using the one minus Pearson correlation method. Genes associated with the indicated GO terms are color-coded according to the legend.

**Figure 5 ijms-25-05579-f005:**
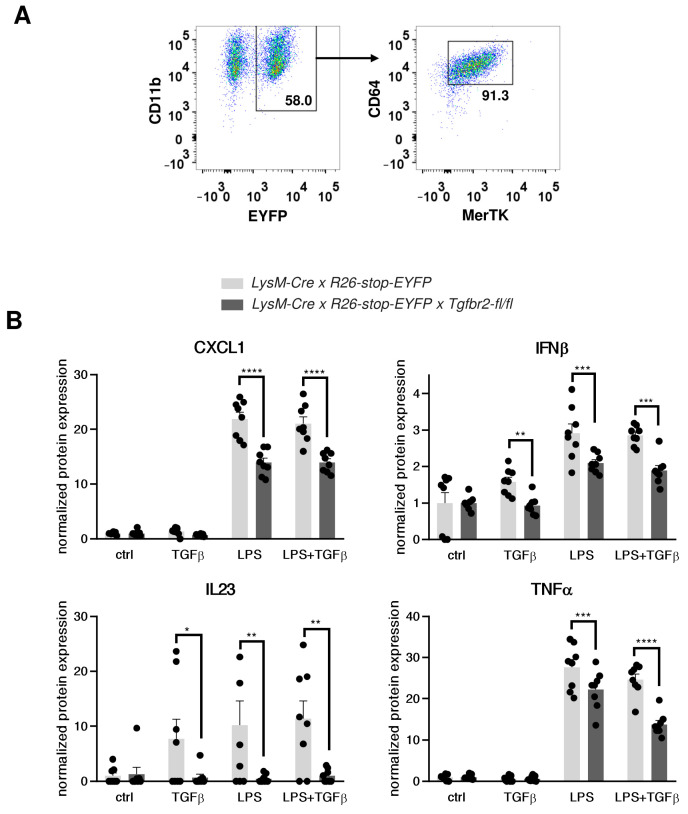
Cytokine production by bone marrow-derived macrophages. Panel (**A**) shows the gating strategy for sorting EYFP^+^CD11b^+^CD64^+^MerTK^+^ bone marrow-derived macrophages. In Panel (**B**), normalized levels of CXCL1, IFN-β, IL-23, and TNF-α produced by bone marrow-derived macrophages stimulated with LPS and/or TGF-β for 24 h are presented. Each dot represents data for an independent experiment, and the bars show mean values ± SEM. The *p*-values were calculated using the unpaired Student’s *t*-test. * *p* < 0.05, ** *p* < 0.01, *** *p* < 0.001, **** *p* < 0.0001.

## Data Availability

The original data presented in the study are openly available in the Gene Expression Omnibus (GEO) database, at https://www.ncbi.nlm.nih.gov/geo/query/acc.cgi?acc=GSE262658.

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
