# Peer review of "TGF-β Signalling Regulates Cytokine Production in Inflammatory Cardiac Macrophages during Experimental Autoimmune Myocarditis"

_ijms, 2024, doi:10.3390/ijms25115579_

Round 1

Reviewer 1 Report

Comments and Suggestions for Authors

In the original article 'TGF-ß signalling regulates cytokine production in inflammatory cardiac macrophages during experimental autoimmune myocarditis' submitted by Tkacz and coworkers to IJMS, the authors have investigated fibrotic remodelling in a in autoimmune myocarditis in a mouse model.

The topic of this manuscript is highly interesting and the manuscript is well written. However, I suggest some points, which should be optimized in a revised manuscript:

1.) The authors specify DCM. However, other cardiomyopathies like ARVC (arrhythmogenic right ventricular cardiomyopathy), can be also caused by myocarditis and cardiac inflammation. Therefore, I suggest to write the manuscript broader instead of focussing on DCM. 

2.) In addition, cardiac inflammation and myocarditis can be also caused by genetic mutations. For example, it is well known, that cardiac inflammation is present in the DSC2 (desmocollin-2) transgenic mouse which leads also to a severe fibrotic remodelling (Brodehl A et al. 2017, PLOS one). I would explain this in the introduction, especially since you have used a genetic mouse model for this study.

3.) Figure 2B. The trichrome staining images are a little bit unsharp and should be optimized. 

3.) The labelling of the figures needs correction. You have labelled two figures with Figure 2.

3.) Material and Methods:

a) How long do you have fixed the tissue? Please specify the time.

b) Please specify the objective lens of the Olympus microscope.

c) Please indicate the concentration of propidium iodide and indicate the details to the washing steps.

d) Please explain in more detail how the heat map was generated. This is unclear to the reader.

e) Please explain, when you used parametric or non-parametric statistical tests for analysis. This is not explained and could not be followed.

In summary, I suggest a major revision for this manuscript and I would be interested to re-review a revised version of this interesting manuscript.

Reviewer 2 Report

Comments and Suggestions for Authors

Dear authors,

I have read the presented article very carefully and I would like to formulate the following observations:

-For a better follow-up of the working hypothesis, I suggest to the authors that the information presented in the introduction (lines 52-84) be accompanied by an image in which the role of macrophages and the involvement of TGF-beta in the inflammatory/myocarditis processes is easily visible.

- Did the authors only make qualitative determinations? In my opinion, the results and discussions also require quantitative interpretation, in relation to the presented figures.

Based on these observations, my recommendation is to publish with major revisions, after the correction of the observations/recommendations made.

Reviewer 3 Report

Comments and Suggestions for Authors

Comments: 

The objective of the study is commendable; dissecting the role of TGF-β signaling specifically in myeloid cells during myocarditis fills a gap in our understanding of the inflammatory processes in cardiac diseases. The use of a conditional knockout model is appropriate for the specific disruption of TGF-β signaling in myeloid cells.

Genetic Model and Efficiency: While the genetic model used is powerful, the variable efficiency of the LysM-Cre recombinase, especially on the BALB/c background, is a significant limitation. This variable recombination efficiency could mean that not all myeloid cells are devoid of TGF-β signaling, potentially diluting the observed effects. I recommend quantifying the efficiency of recombination in the relevant tissues and, if possible, supplementing the study with a more consistent Cre driver or a different genetic background to validate the findings.

Inflammatory and Fibrotic Response Assessment: The findings that TGF-β signaling in myeloid cells does not impact the progression of myocarditis or cardiac fibrosis are intriguing and somewhat counterintuitive given the known roles of TGF-β. The authors should consider whether the timing of the observations might miss later fibrotic developments or resolutions of inflammation. Longitudinal studies could provide additional insight.

Cytokine Analysis: The reduction in pro-inflammatory cytokines in TGF-β signaling-deficient macrophages is an interesting result. However, it raises questions about compensatory mechanisms that might be in play. Are other cell types upregulating cytokine production to compensate for the deficiency in myeloid cells? The study could be strengthened by a broader analysis of the cytokine milieu, including serum cytokine levels and the cytokine-producing capacity of other immune cells.

Macrophage Phenotyping: The study's conclusion that there is no clear TGF-β-dependent shift towards an anti-inflammatory phenotype warrants deeper investigation into macrophage dynamics. Flow cytometry and RNA sequencing data provide a snapshot in time, but they do not capture the dynamic nature of macrophage polarization. Time-course studies analyzing the phenotypes of macrophages at multiple stages of myocarditis would be informative.

Role of Other Immune Cells: The discussion rightly points out the potential role of CD4+ T cells in the disease process. Further experiments using cell depletion or cell transfer techniques could elucidate the roles and interactions of various immune cells in the EAM model. Methodological Details: The methods are well-described, allowing for reproducibility. The use of ELISA to quantify cytokine production is appropriate, and the RNA sequencing approach is robust. Nonetheless, a more detailed explanation of how the differential expression thresholds were determined would be useful. The article could also benefit from an explanation of the bioinformatics pipeline used for RNAseq data analysis.

Broader Implications: The authors should discuss how these findings translate to human myocarditis. While murine models provide valuable insights, there are known differences in immune responses between mice and humans. Any translational potential or limitations should be addressed. Conclusion: Overall, the study is methodologically sound and adds to our understanding of TGF-β's role in cardiac biology. The findings challenge some established views and open up new avenues for research. Future studies could expand on these findings by exploring other cell types, employing longitudinal analyses, and moving towards translational research.

Reviewer 4 Report

Comments and Suggestions for Authors

Greetings to the authors, my comments are as follows. 

Lines 53 to 61 are repetitive and seem to run circles around the same idea.

Line 219 and onward, it is unclear how many mice were used, also the authors mention a minimum of 10 generations, is there a minimum and a maximum ?

Another issue is that throughout the manuscript, the role of the study is explained rather vaguely in the methods section, the authors should describe the main aim of this manuscript in a detailed and clear manner, especially given the rather contradictory conclusions. 

The authors state "While TGF-β did regulate the ex-180 pression of pro-inflammatory genes in our model, we did not observe a clear TGF-β-de-181 pendent shift towards an anti-inflammatory M2 phenotype. This observation could be at-182 tributed to the predominant pro-inflammatory signaling in the inflamed cardiac tissue at 183 this stage, suggesting that the inflammatory macrophages may not have initiated the tran-184 sition from M1 to M2 phenotype yet." -Isn't this paragraph here in contradiction with the general idea of the manuscript that TGF B also downregulates cytokine production ?

Comments on the Quality of English Language

English is fine overall. 

Round 2

Reviewer 1 Report

Comments and Suggestions for Authors

Congratulations. The authors have addressed all my comments in a convincing style. Therefore, I suggest to accept this manuscript for publication in IJMS. 

Author Response

Thank you very much for reviewing our manuscript. In the revised version, minor grammar corrections have been made.

Reviewer 2 Report

Comments and Suggestions for Authors

Dear Authors,

thank you for considering my observations. In this, moment I consider that your article could be published in this journal as it is. 

Author Response

(The authors gave the same response as above.)

Reviewer 3 Report

Comments and Suggestions for Authors

I am satisfied by the author's answers to my queries.

Author Response

(The authors gave the same response as above.)

Reviewer 4 Report

Comments and Suggestions for Authors

The manuscript has improved overall and is interesting and provides a certain level of intricacy. 

Comments on the Quality of English Language

English is fine overall. 

Author Response

(The authors gave the same response as above.)
